# Associations of Clusters of Cardiovascular Risk Factors with Insulin Resistance and Β-Cell Functioning in a Working-Age Diabetic-Free Population in Kazakhstan

**DOI:** 10.3390/ijerph20053918

**Published:** 2023-02-22

**Authors:** Yerbolat Saruarov, Gulnaz Nuskabayeva, Mehmet Ziya Gencer, Karlygash Sadykova, Mira Zhunissova, Ugilzhan Tatykayeva, Elmira Iskandirova, Gulmira Sarsenova, Aigul Durmanova, Abduzhappar Gaipov, Kuralay Atageldiyeva, Antonio Sarría-Santamera

**Affiliations:** 1Department of Special Clinical Disciplines, Faculty of Medicine, Khoja Akhmet Yassawi International Kazakh-Turkish University, Turkistan 161200, Kazakhstan; 2Department of Human Pathology and Physiology, Faculty of Dentistry, Khoja Akhmet Yassawi International Kazakh-Turkish University, Turkistan 161200, Kazakhstan; 3Department of Therapy, Shymkent Medical Institute, Khoja Akhmet Yassawi International Kazakh-Turkish University, Shymkent 160019, Kazakhstan; 4Academic Department of Internal Medicine, University Medical Center, Astana 020000, Kazakhstan; 5Department of Medicine, Nazarbayev University School of Medicine, Astana 020000, Kazakhstan

**Keywords:** cardiovascular risk factors, type 2 diabetes mellitus, insulin resistance, β-cell dysfunction, Kazakhstan

## Abstract

Cardiovascular risk factors aggregate in determined individuals. Patients with Type 2 diabetes mellitus (T2DM) have higher cardiovascular This study aimed to investigate insulinresistance (IR) and β-cell function using the homeostasis model assessment (HOMA) indexes in a general Kazakh population and determine the effect he effect that cardiovascular factors may have on those indexes. We conducted a cross-sectional study among employees of the Khoja Akhmet Yassawi International Kazakh-Turkish University (Turkistan, Kazakhstan) aged between 27 and 69 years. Sociodemographic variables, anthropometric measurements (body mass, height, waist circumference, hip circumference), and blood pressure were obtained. Fasting blood samples were collected to measure insulin, glucose, total cholesterol (TC), triglycerides (TG), and high- (HDL) andlow-density lipoprotein (LDL) levels. Oral glucose tolerance tests were performed. Hierarchical and K-means cluster analyses were obtained. The final sample was composed of 427 participants. Spearmen correlation analysis showed that cardiovascular parameters were statistically associated with HOMA-β (*p* < 0.001) and not with HOMA IR. Participants were aggregated into the three clusters where the cluster with a higher age and cardiovascular risk revealed deficient β-cell functioning, but not IR (*p* < 0.000 and *p* = 0.982). Common and easy to obtain biochemical and anthropometric measurements capturing relevant cardiovascular risk factors have been demonstrated to be associated with significant deficiency in insulin secretion. Although further longitudinal studies of the incidence of T2DM are needed, this study highlights that cardiovascular profiling has a significant role not just for risk stratification of patients for cardiovascular prevention but also for targeted vigilant glucose monitoring.

## 1. Introduction

Cardiovascular risk factors cluster and aggregate within individuals [1]. Clustering of risk factors has been associated with a higher risk of cardiovascular disease. Those risk factors, high blood pressure, abnormal cholesterol [2], high triglycerides [3,4], obesity, lack of physical activity [5], or smoking [6] have also been identified to be associated with a higher incidence of Type 2 diabetes mellitus (T2DM) [7]. Patients with T2DM have a high prevalence of prior higher cardiovascular risk [8].

Diabetes mellitus (DM) incidence is growing globally [9] as well as in Kazakhstan [10]. Diabetes is a complex and heterogeneous disease, more complex than the classification in Type 1 and Type 2 suggest [11]. Biological and clinical implications of putative subtypes of DM require further investigation [12]. Recent novel classification based on clusters attempts to make a refined classification of adult-onset diabetes subgroups and their association with a specific risk of complications, with the aim to provide a useful tool for individualized treatment [13]. Progression differences and complication incidences that are linked with differences in DM subtypes have also been explored to determine possible subtypes of patients that are at risk of developing diabetes [14,15].

Although insulin resistance (IR) and pancreatic β-cell dysfunction are the fundamental features in the development of Type 2 diabetes (T2DM), the pathogenesis of T2DM is still unclear. Both peripheral IR and insufficient insulin release from pancreatic islet β-cells induce hyperglycemia and, therefore, increase insulin demand.

IR may be defined as a subnormal glucose response to endogenous and/or exogenous insulin. It most commonly occurs in association with obesity but may result from multiple other underlying causes, both cell-extrinsic factors. This includes circulating or paracrine molecules (such as hormones, cytokines, lipids, and metabolites) that are released from a cell or tissue other than the target cell/tissue, or absorbed by the intestine from the diet or microbiome action, and cell-intrinsic factors that are most likely due to genetic or epigenetic effects, but may or may not be in the insulin signaling pathway itself [16].

The insulin receptor is a transmembrane protein that is part of the RTK (receptors of tyrosine kinase), which exists as covalently bound receptor dimers at the surface of molecules. This receptor plays crucial roles in all the important functions of cell growth and its metabolism, as well as being related to DM, and thus has been considered a novel therapeutic target. An in-depth analysis of the insulin receptor would help develop an understanding of the regulation of cellular pathways and contribute to the development of novel drugs for T2DM [17].

However, the role and sequence of those inherently complex processes, IR and β-cell dysfunction, and their interrelation for triggering the pathogenesis of T2DM are also undefined [18]. Understanding how these multi-layered molecular networks modulate insulin action and metabolism in different tissues will open new avenues for therapy and prevention of T2DM.

The homeostasis model assessment (HOMA) is derived from a mathematical assessment of the balance between hepatic glucose output and insulin secretion from fasting levels of glucose and insulin [19]. HOMA indexes provide valid estimates of insulin resistance (HOMA-IR) and of β-cell function (HOMA-β). The HOMA index calculation requires only a single measurement of insulin and fasting glucose and is thus considered a valid alternative. Well-conducted prospective studies have determined the predictive validity of both measures to identify patients that are at risk of T2DM developing [20,21,22,23,24].

Patients’ ethnic backgrounds have been associated with differences in the incidence and progression of T2DM [25]. The significant contribution in Asian populations of β-cell dysfunction in the incidence of T2DM, compared to Caucasians is becoming recognized. These pathophysiological differences may have an important impact on therapeutic approaches [26]. Asians may have especially vulnerable β-cells, despite relatively good insulin sensitivity, and be unable to increase insulin secretion further if there is even a slight decrease in insulin sensitivity [27]. Kazakhstan is an ethnically diverse Central Asian country, and its genetic characteristics may hold an intermediate position between European and Eastern Asian populations [28]. In Turkistan, and quite different from other regions of the country, the second most frequent ethnic group after ethnic Kazakhs are Uzbeks [29], with whom Kazakhs possibly share more genetic similarities than ethnic Russians, who in the rest of the country are the second most frequent ethnic group. A previous study has shown that the South Kazakhstan region, where Turkistan belongs to, had the highest proportion of undiagnosed diabetes cases [11].

The objective of this study is to investigate IR and β-cell function in a general Kazakh population and determine the effect that cardiovascular factors may have on those indexes.

## 2. Materials and Methods

The study was conducted at the Clinical Diagnostic Center of the Khoja Akhmet Yassawi International Kazakh-Turkish University (Turkistan, Kazakhstan) between 2019 and 2020. The study population consisted of employees of the Khoja Akhmet Yassawi International Kazakh-Turkish University. The inclusion criteria were age between 27 and 69 years and written informed consent to participate in the study. The exclusion criteria were the presence of already diagnosed kidney disease or diabetes or who were diagnosed with diabetes with the blood tests that were analyzed in this work.

Data on study participants were collected in a patient survey card that contained a summary of the study, a written voluntary informed consent form, passport, and demographic data, questionnaires on lifestyles, as well as anthropometric and laboratory studies.

The Fagerstrom test was used as a questionnaire to determine smoking status, and the Alcohol Use Disorders Identification Test (AUDIT) questionnaire was used to identify the alcohol consumption information. An anthropometric study was conducted for determining the height, and weight for which BMI was calculated. Height was measured by a stadiometer, in which the study participants stood straight, without outerwear and shoes, heels, buttocks, and shoulders were in contact with the vertical plane of the stadiometer. The patients’ heads were kept in the “Frankfurt plane” where the lower boundaries of the orbits were in the same horizontal plane as the external auditory space. When holding their breath on inspiration, the stadiometer plate was lowered to the head of the patient, after which the subject departed. After taking three measurements, the average growth index was determined with an accuracy of 0.1 cm. Body weight was measured on electronic scales. After turning on the scale display to check the performance, when 0.00 g appeared, the participants were asked to stand on the scale. At the same time, shoes, outerwear, and heavy items in pockets (mobile phones, wallets, etc.) were removed. Study participants stood in the center of the scales with their arms freely at their sides. At the same time, the patients looked straight and remained motionless. After three measurements, the mean body weight was recorded to the nearest 0.1 kg. Based on the results of measuring height and body weight, BMI was determined by the formula: weight (kg)/height in m^2^. Waist circumference (WC) was measured while standing, using a soft centimeter tape with an accuracy of 0.1 cm. WC was measured after normal expiration in the middle between the lower rib and the upper part of the iliac crest. According to the measurement of WC, the presence of abdominal obesity (AO) was determined according to the International Diabetes Federation criterion (2005). A WC of more than 94 cm in men and 80 cm in women was taken as AO. Measurement of hip circumference (HC) was carried out with a centimeter tape, in the standing position, on the most protruding part of the gluteal region above the large trochanters, the result was determined with an accuracy of 0.1 cm.

Laboratory methods included the determination of fasting glucose levels, after a 2-h oral glucose tolerance test (OGTT), triglycerides (TG), total cholesterol (TC), high-density lipoprotein (HDL), and low-density lipoprotein (LDL). Blood sampling was carried out from the cubital vein after a 12-h fast. OGTT was performed with 75 g glucose solution, in which the plasma glucose level was measured after 0 and 120 min. For prediabetes, fasting glucose was taken as 6.1–6.9 mmol/L, after OGTT—7.8–11.1 mmol/L (WHO). Biochemical studies were determined in a biochemical analyzer Cobas Integra-400 from Roche (Basel, Switzerland). The listed laboratory studies were carried out in the laboratory of the Clinical Diagnostic Center of Khoja Akhmet Yassawi International Kazakh-Turkish University.

HOMA-IR and HOMA-β were calculated and divided into terciles and in 2 categories, namely IR and Poor β-cell function [30]. HOMA models were calculated as HOMA-IR = fasting insulin (lU/mL) × fasting glucose (mmol/L)]/22.5, and HOMA-β = [20 × fasting insulin (lU/mL)]/[fasting glucose (mmol/L) − 3.5]. IR was defined as values HOMA_IR ≥ 2.5 and Poor β-cell function when HOMA-β ≤ 50.

Correlation analysis was conducted to analyze the possible associations between the different cardiovascular risk factors and glucose metabolism variables.

Cluster analyses were conducted to identify individuals with aggregation of cardiovascular risk factors: hierarchical and k-means. To visualize clustering of individuals, first hierarchical analysis with the Wald method was obtained to create a dendrogram to visually determine the reasonable number of clusters. Second, K-means clusters were finally developed to find the separation of cases based on cardiovascular risk factors as descriptors. SPSS 29.0 statistical software was used for the analyses.

This study was approved by the Commission on Clinical Ethics of the Faculty of Medicine of Khoja Akhmet Yassawi International Kazakh-Turkish University. Before attending the study, the participants were provided with personal explanations regarding the purpose and method of the study, as well as information regarding the processing of the results. Written consent was given by all participants.

## 3. Results

Data were initially available from 632 participants, but data to calculate HOMA-IR and HOMA-β were available only for 488 participants. Cases with fasting blood glucose or OGTT compatible with a diagnosis of diabetes were eliminated. The total sample was composed of 427 subjects. The basal characteristics of cases are depicted in Table 1. The high obesity prevalence and elevated BMI is of note.

As the variables did not show normal distributions, Spearman non-parametric correlation analysis was conducted. Table 2 shows the Spearman correlation between the different quantitative variables related either to cardiovascular risk factors or glucose metabolism. All the cardiovascular parameters were inversely statistically associated with HOMA-β and none of them with HOMA IR.

Figure 1 shows the dendrogram of hierarchical clustering demonstrating that the separation of subjects into three clusters is well depicted.

Table 3 reflects the values of cardiovascular risk factors of the three proposed clusters created using the K-means method. Table 4 reveals the glucose metabolism characteristics of participants that were aggregated into the three clusters. Significant associations were identified for beta-cell functioning, but not for IR.

Appendix A show the distribution of cardiovascular risk factors by HOMA terciles as well as IR and Poor β-cell functioning.

## 4. Discussion

The findings of this study, combining different analytical methods to identify the possible relationship between cardiovascular risk factors and homeostasis indexes that reflect susceptibility to T2DM, suggest the existence of a significant association between cardio-metabolic alterations and β-cell function, as measured by HOMA-β, while not such an association with IR. Age also showed a strong independent effect on β-cell dysfunction. Another relevant finding of this study is the aggregation of cardiovascular risk factors in certain groups of this population as well as the association of higher cardiovascular risk with age and with β-cell deficiency. Is also relevant to mention the elevated proportion of overweight and obese participants as well as abdominal obesity in this population.

HOMA-IR and HOMA-β are widely accepted surrogate measures of IR and β-cell dysfunction in clinical and epidemiological studies [31], but the interpretation and extrapolation of the current findings for its application for clinical practice or public health decision-making should be cautious; showing more or less deteriorated glucose homeostasis indexes reflecting either IR or poor β-cell functioning should not be immediately associated with a higher risk of incidence of T2DM.

The most common understanding of the T2DM pathogenic process is that IR is the primary glucose homeostasis abnormality, with β-cell dysfunction being a later manifestation when β-cells no longer sustain sufficient insulin secretion and became ‘exhausted’. However, “primary” β-cell dysfunction as an independent abnormality in the early phases of the development of dysglycemia has also been suggested [32,33].

Different mechanisms (glucotoxicity, lipotoxicity, oxidative stress, endoplasmic reticulum stress, inflammatory stress, amyloid formation, or decreased incretins) have been suggested for β-cell death [34,35]. The coexistence of adverse cardiovascular risk profiles, including overweight and obesity, high blood pressure levels, and lipid alterations in certain individuals, may create a “hostile” metabolic environment that, when acting in concert with age, may be associated with those factors and reduce functional β-cell mass and increase the risk for T2DM [36,37].

Cluster 1 that was identified in this study included 37% of the sample that were analyzed and revealed advanced age and the worst cardiovascular risk in terms of blood pressure and lipid profiles, a high prevalence of obesity, and a significantly poorer β-cell function. In contrast, the cluster with lower age and most favorable cardiovascular risk showed the best pancreatic β-cell function. These data did not show an association between cardiovascular risk factors and IR.

Obesity has been classically considered a hallmark of IR [38]. We did not find this association in this study. In obesity, adipose tissue releases increased amounts of non-esterified fatty acids, glycerol, hormones, pro-inflammatory cytokines, and other factors that are involved in the development of IR [39]. However, it is only when IR is accompanied by dysfunction and failure of pancreatic β-cells to control blood glucose levels that this results in T2DM.

Our results indicate an association between obesity and reduced β-cell function [40]. Obesity may be linked to pancreatic fat infiltration leading to impaired β-cell function, and the development of T2DM [41,42]. Excess cholesterol may have a direct pancreatic β-cell lipotoxicity, contributing as an underlying factor in the progression of T2DM [43]. Cholesterol is important for β-cell function and survival, but it can cause β-cell loss if allowed to accumulate in the cells in an unregulated manner [44]. Cholesterol excess impacts several steps of the metabolic machinery that are involved in glucose-stimulated insulin release localized at the endoplasmic reticulum, mitochondria, and the cell membrane [45,46]. This study adds to the growing body of literature that suggests that obesity and lipid alteration contribute to β-cell dysfunction [47].

Aging is one of the most important factors that is implicated in the major changes that are associated with deteriorated glucose metabolism through β-cell function [48,49], and appears to be independent of IR, BMI, and waist circumference [50]. The cause of this age-dependent functional decline is not known [51]. It is also not known whether this effect is mediated by a reduction in incretin secretion or not or may be associated with an aging-related β-cell resistance to the incretin effect, thus needing an increased release of incretin hormones, glucagon-like peptide-1 and gastric inhibitory polypeptide, to stimulate adequate insulin secretion in response to the glucose load [52,53]. A better understanding of all the factors that alter the proper regulation of glucose metabolism at advanced ages will facilitate the design of therapies that allow for better management of glycemia [54].

The study has some limitations. First, this is a selected working-age population from one company. The generalizability of these data may be limited. The cross-sectional nature of the study design prevents establishing causality in the direction of the associations identified. Cut-off points for HOMA-IR and HOMA-β are not standardized; other cut-off points may have rendered different results. A lack of a standardized universal insulin assays limit their use for routine assessment of insulin resistance in the clinical setting and may have affected our results. This analysis has separated the cases into three clusters but having determined another number of clusters may have rendered different results. The same limitation applies for our cut-off point of IR or poor-β-cell function. Also, the ethnic or genetic factors that may influence the glucose homeostasis indexes may be valid only for specific populations where they have been obtained. No data regarding two relevant variables, use of drugs or physical activity were available for these analyses. Lastly, this study does not aim to provide mechanistic explanations of the possible associations that may have been identified by analyzing these data.

## 5. Conclusions

T2DM is a complex and multifactorial global health problem that affects millions of people worldwide, has a significant impact on their quality of life, and results in grave consequences for healthcare systems. In T2DM, deficiency of β-cell function, primary or secondary to peripherally developed IR, is a paramount factor leading to dysregulated blood glucose and long-lasting hyperglycemia. The results from this work indicate that common and easy-to-obtain biochemical and anthropometric measurements capturing relevant cardiovascular risk factors are associated with significant β-cell-deficient insulin secretion. Although further longitudinal studies of the incidence of T2DM are needed, this study highlights that cardiovascular profiling has a significant role, not just for risk stratification of patients for cardiovascular prevention, but also for targeted vigilant glucose monitoring.

## Figures and Tables

**Figure 1 ijerph-20-03918-f001:**
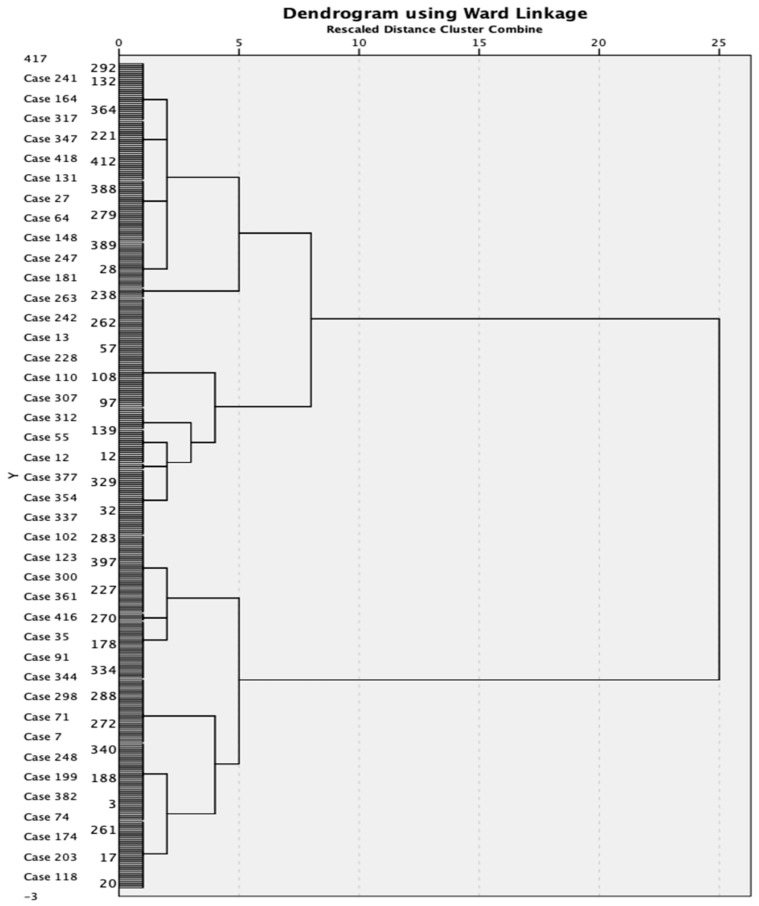
Dendrogram of hierarchical clustering.

**Table 1 ijerph-20-03918-t001:** General characteristics of the analyzed sample.

		Frequency	Percent
Sex	Men	125	29.3
	Women	302	70.7
Age groups	20–29	8	1.9
	30–39	100	23.5
	40–49	106	25.1
	50–59	125	29.1
	60–69	88	20.5
Kazakh		376	88.1
Other ethnicity		51	11.9
Smoking		53	12.3
Alcohol intake		119	27.9
BMI	Normal	132	30.9
	Overweight	144	33.7
	Obesity	151	35.3
Abdominal Obesity		285	66.7
IR		98	23.0
Poor β-cell function		111	25.8
Total		427	
		Median	Minimum-Maximum
Age		48.00	28–69
BMI		27.91	14.56–48.00
Waist circumference		94	55–131
Hip circumference		104	37–146
SBP		120	60–180
DBP		80	50–100
TC		4.85	1.46–8.40
LDL		2.17	0.28–5.51
HDL		1.21	0.28–4.21
TG		2.03	0.72
Fasting blood sugar test		5.45	0.76
Oral glucose tolerance test		5.67	1.13
Insulin		8.01	3.89
HOMA-IR		1.76	1.54–6.70
HOMA-β		82.08	17.51–480.57

IR: insulin resistance, BMI: body mass index, SBP: systolic blood pressure, DBP: diastolic blood pressure, TC: total cholesterol, HDL: high density lipoprotein cholesterol, LDL: low density lipoprotein cholesterol, TG: triglycerides.

**Table 2 ijerph-20-03918-t002:** Spearman correlation coefficients of cardiovascular factors with glucose metabolism factors.

	HOMA IR	HOMA-β	Fasting Blood Sugar	Oral Glucose Tolerance Test	Insulin
HOMA_IR	10.000	0.552 **	0.213 **	−0.031	0.955 **
HOMA_β	0.552 **	10.000	−0.661 **	−0.266 **	0.758 **
Fasting blood sugar	0.213 **	−0.661 **	10.000	0.322 **	−0.062
Oral glucose tolerance test	−00.031	−00.266 **	0.322 **	10.000	−0.119 *
Insulin	0.955 **	0.758 **	−00.062	−0.119 *	10.000
Age	−0.121 *	−0.444 **	0.426 **	0.247 **	−0.250 **
SPB	−0.020	−0.287 **	0.343 **	0.297 **	−0.119 *
DBP	−0.001	−0.265 **	0.328 **	0.266 **	−0.096 *
BMI	0.040	−0.265 **	0.361 **	0.236 **	−0.063
Waist circumference	0.037	−0.280 **	0.367 **	0.243 **	−0.072
Hip circumference	0.030	−0.264 **	0.358 **	0.222 **	−0.074
TC	−0.013	−0.231 **	0.290 **	0.324 **	−0.096 *
DL	−0.058	−0.223 **	0.232 **	0.184 **	−0.122 *
HDL	0.023	0.166 **	−0.186 **	−0.081	0.078
TG	0.023	−0.104 *	0.137 **	0.217 **	−0.022

* *p* < 0.05; ** *p* < 0.001. BMI: body mass index, SBP: systolic blood pressure, DBP: diastolic blood pressure, TC: total cholesterol, HDL: high density lipoprotein cholesterol, LDL: low density lipoprotein cholesterol, TG: triglycerides.

**Table 3 ijerph-20-03918-t003:** Cardiovascular risk factor distribution of the three clusters based on K-means.

	Cluster		
	1	2	3
Number of cases	154	74	186
Age	57	47	43
BMI	32.80	26.86	25.41
Waist circumference	104	86	85
Hip circumference	113	100	100
SBP	145	127	106
DBP	90	84	71
TC	5.14	4.81	4.61
LDL	2.38	2.17	2.04
HDL	1.19	1.35	1.30
TG	2.02	1.84	1.84

BMI: body mass index, SBP: systolic blood pressure, DBP: diastolic blood pressure, TC: total cholesterol, HDL: high density lipoprotein cholesterol, LDL: low density lipoprotein cholesterol, TG: triglycerides.

**Table 4 ijerph-20-03918-t004:** Glucose metabolism characteristics of the three clusters based on k-means of cardiovascular risk factors: Kruskal–Wallis and chi-square tests for medians and percentages.

		Clusters			
Median		1	2	3	*p*-Value
Fasting Blood Sugar		5.90	5.37	5.20	0.000
Oral Glucose Tolerance Test		5.80	5.60	5.20	0.000
Insulin		7.35	7.75	7.92	0.064
HOMA IR		1.77	1.85	1.75	0.982
HOMA-β		59.06	86.36	9.35	0.000
Percentages					
IR		22.70%	21.30%	23.90%	0.897
Poor β-cell function		40.30%	18.70%	16.50%	0.000
HOMA-IR Tertiles	1	39.7%	18.4%	41.9%	
	2	32.4%	16.9%	50.7%	
	3	38.8%	18.7%	42.4%	0.580
HOMA-β Tertiles	1	54.7%	12.9%	32.4%	
	2	36.4%	22.9%	40.7%	
	3	19.6%	18.1%	62.3%	0.000

IR: insulin resistance.

## Data Availability

Due to ethical requirements, data are not available.

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
