# Peer review of "Associations of Clusters of Cardiovascular Risk Factors with Insulin Resistance and Β-Cell Functioning in a Working-Age Diabetic-Free Population in Kazakhstan"

_ijerph, 2023, doi:10.3390/ijerph20053918_

Round 1

Reviewer 1 Report

In the article titled Associations of Clusters of Cardiovascular Risk Factors with Insulin Resistance and Β-cell Functioning in a Working-age Diabetic Free Population in Kazakhstan, Saruarov et al. analyzed the insulin resistance and β-cell function in a general diabetes free Kazakh population. The authors included the appropriate number of participants. The anthropometric measurements are described in great detail, while on the other hand, the description of the calculations of HOMA indexes is deficient. Below I have listed a few of my inquiries and comments on the article.

1.      It is not clear from the text how HOMA indexes were calculated. Therefore, the equations used to calculate HOMA-IR and HOMA-β should be added.

2.      Describe how to interpret the HOMA values. Where are the cut-off values?

3.      Although the authors mentioned some drawbacks of HOMA index, there are several limitations that should be described in more detail.

4.      In table 1, the numbers of participants with “poor beta cell function” and “insulin resistance” are mentioned. How did you determine this? Have you determined this based on the value of HAM-β and HOMA-IR? What were the cut-off values?

5.      In table 3, 3 proposed clusters created using the K-means method are shown. It is not clear from the text which data was used for the hierarchical clustering. This should be explained in more detail. 

Reviewer 2 Report

The manuscript ASSOCIATIONS OF CLUSTERS OF CARDIOVASCULAR RISK FACTORS WITH INSULIN RESISTANCE AND β-CELL FUNCTIONING IN A WORKING-AGE DIABETIC FREE POPULATION IN KAZAKHSTAN by the authors Yerbolat Saruarov et al it is interested, however It is necessary to attend to some corrections that I think are important before accepting the manuscript and then ready

 The manuscript required an extensive English edition for example:

Abstract page 1, line 22, please substitute the verb of the objective in the past tense

Line 29 please add the abbreviations for Triglycerides (TG), Total cholesterol (TC), Insulin (IR) Resistance, also in the abstract add against which the results obtained are compared, this clarification is missing and the significance (p=?).

Page 2, line 46, please add what does CVD mean? Line 47 abbreviation for high blood pressure,

Line 49 The first time a phrase is mentioned that it uses an abbreviation and not after two paragraphs, please correct this. T2DM (line 61).

Please add the abbreviation for insulin resistance (throughout the entire document) lines:64,74,126,167,185,202,203,206,207,221.

Page 2 the section of introduction please add and develop the participation of the insulin receptor in the insulin resistance, this very import.

Page 2 The Section of materials and methods.

The variables that were analyzed may be influenced by the type of drug used by the individuals in the study. Please add this data as well as physical activity if they were male or female.

Line 53, what does DM mean?

Line 57 I think the phrase could be better the progressions difference and complications incidence linked whit ...

Line 73, I think the phrase could be better T2DM developing

Please add in line 79 between 2019-2020, delete in

Page 3, line 110 what does IDF mean?

Line116 please add abbreviations for total cholesterol and triglycerides

Line 142 please add 427 ? (Patients, individual, citizen, subjects).

Page 4 table 1. the total sum of abdominal obesity plus insulin resistance, plus poor b-cell function it is =494 not 427 and the percentage it is 115.5% not 100%. Please review this.

Please in the figure caption add a list of abbreviations

Due to the type of study, the mean or standard deviation is not accepted, but rather the median and maximum and minimum values because the distribution is not parametric, for this reason another statistical test could be applied, for example, the Mann-Whitney Rank Sum Test normalized with Normality Test (Shapiro-Wilk)

Page 5 table 2 please substitute abbreviation the Total cholesterol (TC) tryglicerides (TG) and a list of abbreviations, In addition, the significance is lacking against which groups the comparison is made

Figure 1 dendrogram of hierarchical clustering it is very little and this and this makes it difficult to observe, could you please enlarge it

Table 3 substitute in the table: Systolic Blood pressure, Diastolic Blood pressure, Total cholesterol and triglycerides by abbreviation and add the list at the foot of the figure.

Page 6 lines 185,230,231, 57 I think the phrase could be better T2DM process.

the same case for obesity prevalence, line 199

Line 224, what does GLP-1 and GIP mean?

Page 7 line 233-241, I think this section fits more in the introduction and gives the guideline for the justification of the work and not in the discussion section

Although the authors mention the cut-off points for the indices in the limitations section, these were not standardized, so this is very worrying, since the method must first be standardized and compared with others, so that the values are reliable, because otherwise the results and the discussion are very fragile

The reference 52 not cited in the text.

Thank you for giving me the opportunity to review your work, carefully the reviewer

Round 2

Reviewer 2 Report

Dears authors

Thanks you for answered satisfactorily to the questions of the article with title ASSOCIATIONS OF CLUSTERS OF CARDIOVASCULAR RISK FACTORS WITH INSULIN RESISTANCE AND β-CELL FUNCTIONING IN A WORKING-AGE DIABETIC FREE POPULATION IN KAZAKHSTAN by de authors: Yerbolat Saruarov, Gulnaz Nuskabayeva, Mehmet Ziya Gencer, Karlygash Sadykova, Mira Zhunissova, Ugilzhan Tatykayeva, Elmira Iskandirova, Gulmira Sarsenova, Aigul Durmanova, Abduzhappar Gaipov, Kuralay Atageldiyeva, Antonio Sarría-Santamera. However in the introduction section, you added the definition of insulin resistance and it is correct, but the participation of the insulin receptor in the resistance is not mentioned anywhere in the document and your participation is very important, could you please add something about it. For example The Insulin Receptor: An Important Target for the Development of Novel Medicines and Pesticides Int J Mol Sci.2022 Jul 14;23(14):7793. doi: 10.3390/ijms23147793.
